# Screen for mitochondrial DNA copy number maintenance genes reveals essential role for ATP synthase

Atsushi Fukuoh[1,2,3,†], Giuseppe Cannino[1,†], Mike Gerards[1], Suzanne Buckley[1], Selena Kazancioglu[1], Filippo Scialo[1], Eero Lihavainen[4], Andre Ribeiro[4], Eric Dufour[1] & Howard T Jacobs[1,5,*]

## Abstract

The machinery of mitochondrial DNA (mtDNA) maintenance is only partially characterized and is of wide interest due to its involvement in disease. To identify novel components of this machinery, plus other cellular pathways required for mtDNA viability, we implemented a genome-wide RNAi screen in *Drosophila* S2 cells, assaying for loss of fluorescence of mtDNA nucleoids stained with the DNA-intercalating agent PicoGreen. In addition to previously characterized components of the mtDNA replication and transcription machineries, positives included many proteins of the cytosolic proteasome and ribosome (but not the mitoribosome), three proteins involved in vesicle transport, some other factors involved in mitochondrial biogenesis or nuclear gene expression, > 30 mainly uncharacterized proteins and most subunits of ATP synthase (but no other OXPHOS complex). ATP synthase knockdown precipitated a burst of mitochondrial ROS production, followed by copy number depletion involving increased mitochondrial turnover, not dependent on the canonical autophagy machinery. Our findings will inform future studies of the apparatus and regulation of mtDNA maintenance, and the role of mitochondrial bioenergetics and signaling in modulating mtDNA copy number.

**Keywords** complex V; DNA replication; mitochondrial biogenesis; mitochondrial DNA; mitophagy; nuclease; nucleoid; reactive oxygen species
**Subject Categories** Methods & Resources; Metabolism
**Mol Syst Biol.** (2014) 10: 734

## Introduction

Eukaryotes that use mitochondrial oxidative phosphorylation (OXPHOS) to generate ATP maintain a separate mitochondrial genome (mtDNA), encoding a subset of OXPHOS protein subunits, together with some components of the machinery of intramitochondrial protein synthesis. The maintenance and expression of mtDNA is otherwise dependent on nuclear-coded gene products, constituting a separate apparatus for genome maintenance and gene expression within the cell (McKinney & Oliveira, 2013).

The core machinery of mtDNA replication is broadly conserved among eukaryotes. DNA replication is assumed to depend on the only DNA polymerase consistently found in mitochondria, DNA polymerase γ (PolG; Kaguni, 2004), a member of the family A DNA polymerases (Ito & Braithwaite, 1991). Its closest prokaryotic homologue is the phage T7 DNA polymerase, and it is assumed to function in concert with the mitochondrial helicase Twinkle (Spelbrink *et al*, 2001), a homologue of phage T7 helicase-primase (gp4). Twinkle is absent from yeast, where other helicases are involved in mtDNA replication. Maintenance of mtDNA requires also mitochondrial transcription factors A (mt-TFA or TFAM), needed for mtDNA compaction and transcription (Larsson *et al*, 1998; Kang & Hamasaki, 2005; Campbell *et al*, 2012), and B (mt-TFB2, TFB2M; Matsushima *et al*, 2004). Three other proteins are essential for mtDNA maintenance, namely mtSSB, the mitochondrial single-stranded DNA-binding protein (Maier *et al*, 2001), RNase H1 (Cerritelli *et al*, 2003), and, in some organisms, a second DNA polymerase, PrimPol, that additionally has primase activity (García-Gómez *et al*, 2013). A number of other proteins are required to maintain normal mtDNA copy number or topology in different organisms (Contamine & Picard, 2000; Copeland, 2012). These include the DNA-binding AAA protein ATAD3A (He *et al*, 2007), some enzymes of nucleotide metabolism and transport (Saada, 2004), proteins with roles in mitochondrial membrane dynamics (Jones & Fangman, 1992; Wong *et al*, 2000; Elachouri *et al*, 2011; Vielhaber *et al*, 2013), chaperones (Ciesielski *et al*, 2013), exonucleases (Kornblum *et al*, 2013), proteases (Herlan *et al*, 2003; Matsushima *et al*, 2010; Sesaki *et al*, 2003), and even cytoskeletal proteins (Reyes *et al*, 2011). Mitochondria also contain

1 BioMediTech and Tampere University Hospital, University of Tampere, Tampere, Finland
2 Department of Clinical Chemistry and Laboratory Medicine, Kyushu University Graduate school of Medical Sciences, Fukuoka, Japan
3 Department of Medical Laboratory Science, Junshin Gakuen University, Fukuoka, Japan
4 Department of Signal Processing, Tampere University of Technology, Tampere, Finland
5 Research Program of Molecular Neurology, University of Helsinki, Helsinki, Finland
*Corresponding author. Tel: +358 3 3551 7731, +358 50 341 2894; E-mail: howard.t.jacobs@uta.fi
†These authors equally contributed to this work.

topoisomerases, ligases, and other nucleases, although their specific roles in mtDNA metabolism are unclear. While a crude DNA synthetic machinery can be reconstituted *in vitro* from a minimal set of these proteins, the full complement of proteins required for faithful mtDNA replication *in vivo* remains to be determined.

Some components of the mtDNA maintenance machinery are shared with the nuclear compartment, including RNase H1 (Cerritelli *et al*, 2003) and many proteins implicated in base-excision repair (Alexeyev *et al*, 2013). Mostly, these are synthesized in two or more isoforms routed to different cellular compartments, for example, via differential splicing, alternative translational start sites (Suzuki *et al*, 2010), or ambiguous targeting signals (Karniely & Pines, 2005).

mtDNA is packaged together with TFAM and some other proteins into discrete intramitochondrial structures of variable composition, called nucleoids, by analogy with those of bacteria (Spelbrink, 2010; Bogenhagen, 2012). They contain a number of replication proteins whose functional roles are poorly understood, as well as proteins implicated in other cellular processes, including metabolic enzymes and chaperones, and proteins involved in intramitochondrial protein synthesis (Hensen *et al*, 2014). The apparatus of mitochondrial translation has been functionally implicated in mtDNA maintenance in yeast (Contamine & Picard, 2000), though not in metazoan cells (Storrie & Attardi, 1972).

The importance of mtDNA maintenance for cell physiology and homeostasis is underscored by the finding that its dysfunction leads to diverse types of human disease, including both infantile and late-onset pathologies, showing a bewildering variety of tissue specificities (Shadel, 2008; Rötig & Poulton, 2009; Ylikallio & Suomalainen, 2012). Loss of mitochondrial genome integrity or fidelity is also associated with aging (Oliveira *et al*, 2010; Bratic & Larsson, 2013). Identifying the full set of gene products involved in faithful mtDNA maintenance is thus of broad interest and importance.

To this end, we implemented a genome-wide (blinded) screen of *Drosophila* S2 cells, using dsRNA-based RNA interference (RNAi), taking advantage of the fact that S2 cells tolerate loss of mtDNA and continue to grow within the time scale of a typical experiment, despite decreased OXPHOS capacity. Furthermore, mtDNA nucleoids may be identified in these cells on the basis of fluorescence signal from the topology-dependent DNA-intercalating dye Pico-Green (Ashley *et al*, 2005). Effects on mtDNA copy number were then probed further using quantitative PCR (QPCR), with additional experiments conducted on the cellular phenotypes produced by knockdown of specific genes identified in the screen, notably those encoding subunits of ATP synthase, in order to test aspects of the mechanisms by which they may act.

## Results and Discussion

### Implementation and outcome of the primary screen

We set out to screen a genome-wide *Drosophila* dsRNA library in S2 cells, scoring for disappearance of the PicoGreen signal of mtDNA nucleoids as indicative of genes required for mtDNA maintenance. In initial trials, we found it difficult to pick out the nucleoid signal against background cytoplasmic fluorescence. Using a dsRNA against the PolG catalytic subunit (*tamas*) as a positive control,

and a dsRNA directed against GFP as a negative control, we established a protocol whereby it was possible reliably to score (by eye) the disappearance of nucleoid signal (see Fig 1). This involved applying the test dsRNA for 5 days in a 96-well plate format, with addition on day 3 of a dsRNA directed against TFAM. Although prolonged incubation with TFAM dsRNA itself led to mtDNA depletion, the shorter-term treatment conversely enhanced the nucleoid signal in negative control cells, whereas it was decreased to very low levels in cells treated with the positive control dsRNA.

The primary screen, conducted blind, was successful in identifying as positives most of the known factors involved in mtDNA metabolism (Table 1, category 1), giving confidence in its validity. In total, 105 dsRNA targets were initially judged as positive (Supplementary Table S1), of which almost half were recorded also as leading to cell death in a fraction of the cells. Consistent with previous studies (Rämet *et al*, 2002; Boutros *et al*, 2004), a further 276 targets (Supplementary Tables S1 and S2) gave massive cell death but no specific loss of nucleoid signal and were considered to represent essential genes that could not be studied further. Finally, an additional 132 targets were judged to give an abnormal outcome without complete loss of PicoGreen nucleoid signal (Supplementary Tables S1 and S3), but were not analyzed further, although many fell into similar categories or pathways as those on the positives list. The specificity and knockdown efficiency of this procedure has previously been documented (Clemens *et al*, 2000; Kleino *et al*, 2005). Knockdown was here verified at the RNA level by qRT-PCR for 17 specific targets (see SI).

We compared subjective judgment against a computational method to measure punctate fluorescence intensity (details to be published elsewhere). The latter gave many false positives due to variable background fluorescence, as well as false negatives due to cell debris. Based on rescreening, we judged the manual method to be superior (see SI for details), and we set criteria for defining positives as described below.

### Rescreening to identify definitive positives

Positives were considered as confirmed if three positive but no negative findings were obtained. Those where a negative or ambiguous finding was recorded were retained only where three times as many clearly positive findings were obtained upon exhaustive rescreening (double asterisks in column H of Table 1) otherwise they were considered as false positives (double asterisks in column G of Supplementary Table S4).

Five targets that did not give consistently positive findings during rescreening were noted to give rise to multiple splice variants, encoding at least one polypeptide predicted to be mitochondrially localized. For these, we tested dsRNAs targeted specifically on the relevant splice variants, confirming several additional positives (Table 1, green background), whereas dsRNAs targeted against other splice variants or the entire gene were judged negative (Supplementary Table S4, blue background).

Of the original 105 positives, 83 were retained, 20 were reassigned as negative, and one was reassigned as abnormal (pink background in Supplementary Table S3). One was discarded because the dsRNA detected a pseudogene of a gene already in the list, another because a revised gene model combined it with another

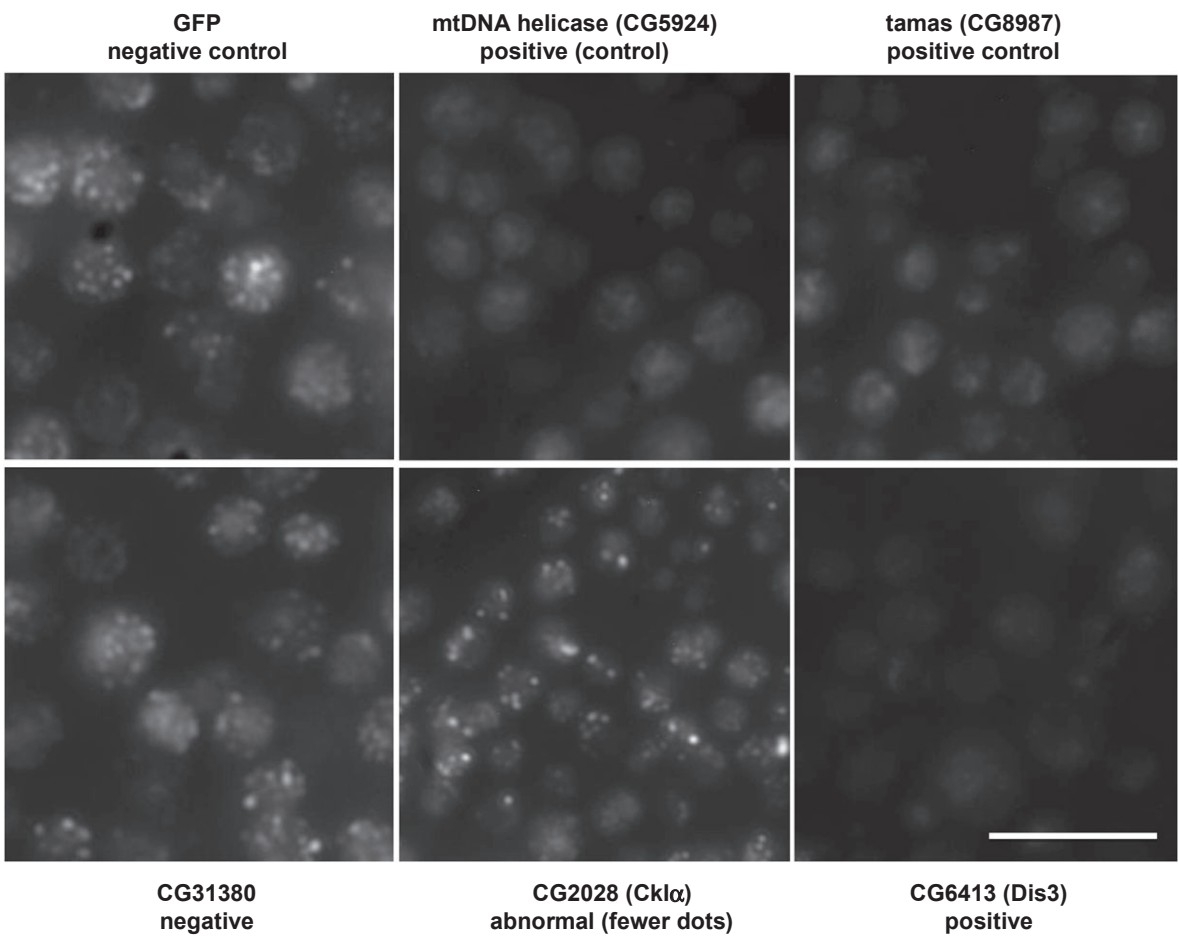

**Figure 1.  Screening of *Drosophila* dsRNA library by PicoGreen nucleoid fluorescence in S2 cells.**
Micrographs of S2 cells stained with PicoGreen, following 5 days of treatment with the dsRNA indicated. GFP and tamas (Polg, CG8987) were used as negative and positive controls, respectively. In rescreening, CG5924 was also used as a positive control. Both were detected in the blinded screen as positives. Other images show a typical negative (CG31380), a typical positive (CG6413) and a typical case of a target classed as abnormal, in this case CG2028 (CkIα), which showed a decreased number of nucleoid signals per cell. Images are optimized for brightness and contrast but with no other manipulations. Scale bar indicates 50 μm.

positive, while another was subsequently re-annotated as two separate genes (but shown as a single entry in Table 1). Positive findings were also obtained for specific splice isoforms of three of the negatives. Thus, the confirmation of 86 out of 106 initial targets indicates a false-positive rate of 18%. One additional positive (CG5794) was unexpectedly identified by a dsRNA nominally targeted against a different gene. The positives fell into seven distinct classes: mitochondrial DNA replication or transcription, cytosolic translation, the proteasome, ATP synthase, mitochondrial dynamics or biogenesis, nuclear gene expression, and a seventh, miscellaneous category.

Although the primary negatives were not rescreened systematically, some that fell into similar functional pathways as definitive positives were re-evaluated using the same criteria. Eleven were promoted to the positives list (yellow background in Table 1), including, for example, most other subunits of ATP synthase. This implies that the initial screen may have missed as many positives as were actually retained, implying a false-negative rate of up to 1%. The overall results of the screen are summarized in Fig 2. The final number of definitive positives was 97, counting only once those with > 1 positive splice variant.

## Copy number of mtDNA

For genes on the definitive positives list, we carried out QPCR to assess changes in mtDNA copy number after 5 days of dsRNA treatment (without concomitant TFAM knockdown), normalized against a single-copy nuclear DNA standard. Based on this assay (see Table 1, column G, raw data in Supplementary Table S5), we classified the positives as showing substantial (++, ≤ 60%) or modest (+) mtDNA depletion, no significant copy number change (0), or an *increase* in mtDNA (−).

Although these classes seem arbitrary, we found that RNAi knockdown of well-characterized components of the mtDNA replication machinery all gave values in the 20-60% range (++). Surprisingly, we identified only 6 new genes from the screen whose knockdown produced a comparably severe mtDNA depletion. Three of these encode subunits of ATP synthase. The others were CG5794, encoding a de-ubiquitinating enzyme with unknown substrate(s), *TweedleY*, previously identified as a cuticular protein (Guan *et al*, 2006), and *pointed*, a well-studied Ets family transcription factor (Klaes *et al*, 1994; Morimoto *et al*, 1996). The fact that knockdown

**Table 1.** Definitive positives

| Category number | Category name | CG number | Official name or symbol | Human orthologue | Other or common name(s) or putative function | mtDNA depletion | > 3 rescreenings | Note |
|---|---|---|---|---|---|---|---|---|
| 1 | mtDNA replication and transcription | CG3910 | mtTFB2 | TFB2M | mt transcription factor B2 | + | * | |
| | | CG4217 | TFAM | TFAM | mtTFA, mt transcription factor A | ++ | | |
| | | CG4337 | mtSSB | SSBP1 | mt single-stranded DNA-binding protein | ++ | | |
| | | CG4644 | mtRNApol | POLRMT | mt RNA polymerase | ++ | | |
| | | CG5924 | CG5924 | PEO1 | Twinkle DNA helicase | ++ | | |
| | | CG6815 | belphegor | ATAD3A, B, C | AAA domain containing protein 3 | 0 | ** | |
| | | CG7175 | mTerf5 | | mt transcription termination factor 5 | ++ | | |
| | | CG8729 | rnh1 | RNASEH1 | Ribonuclease H1 | 0 | ** | Weak positive, diminished nucleoid signal seen in some cells, also some cell death |
| | | CG8987 | tamas | POLG | DNA polymerase γ catalytic subunit | ++ | | |
| | | CG18124 | mTTF | | mt transcription termination factor | ++ | | |
| | | CG33650 | DNApol-γ35 | POLG2 | DNA polymerase γ accessory subunit, CG8969 | + | ** | |
| 2 | Cytosolic translation | CG1821 | RpL31 | RPL31 | Large subunit | – | | |
| | | CG3203 | RpL17 | RPL17 | Large subunit | – | | |
| | | CG3751 | RpS24 | RPS24 | Small subunit | nt | | |
| | | CG3922 | RpS17 | RPS17, RPS17L | Small subunit | – | | |
| | | CG3997 | RpL39 | RPL39, RPL39L | Large subunit | 0 | | |
| | | CG4111 | RpL35 | RPL35 | Large subunit | nt | | |
| | | CG4759 | RpL27 | RPL27 | Large subunit | nt | | |
| | | CG7283 | RpL10Ab | RPL10A | Large subunit | nt | | |
| | | CG7490 | RpLP0 | RPL0 | Large subunit | nt | | |
| | | CG7622 | RpL36 | RPL36 | Large subunit | – | * | |
| | | CG7726 | RpL11 | RPL11 | Large subunit | nt | * | Cell death not substantial in some repeats |
| | | CG8922 | RpS5a | RPS5 | Small subunit | nt | * | |
| | | CG9282 | RpL24 | RPL24 | Large subunit | nt | * | |
| | | CG9677 | Int6 | EIF3E | eIF3, Initiation factor 3, subunit E | nt | * | Abnormal distribution of some residual nucleoid signal within cells |
| | | CG11522 | RpL6 | RPL6 | Large subunit | | | |

**Table 1** (Continued)

| Category number | Category name | CG number | Official name or symbol | Human orthologue | Other or common name(s) or putative function | mtDNA depletion | > 3 rescreenings | Note |
|---|---|---|---|---|---|---|---|---|
| 3 | Proteasome | CG1341 | Rpt1 | PSMC2, 5, 6 | ATPase regulatory subunit 2 (or 5 or 6) | + | | |
| | | CG5266 | Pros25 | PSMA2 | Core α-type subunit 2 | + | ** | |
| | | CG9324 | Pomp | POMP | Proteasome maturation protein | − | | |
| | | CG9327 | Pros29 | PSMA4 | Core α-type subunit 4 | + | * | |
| | | CG10149 | Rpn6 | PSMD11 | Non-ATPase regulatory subunit 11 | + | | |
| | | CG16916 | Rpt3 | PSMC4 | ATPase regulatory subunit 4 | + | | |
| | | CG18174 | Rpn11 | PSMD14 | Non-ATPase regulatory subunit 14 | 0 | ** | Severe cell death: few cells remained after 5 days |
| 4 | ATP synthase | CG2968 | l(1)G0230 | ATP5D | δ Subunit of F1 (stalk) | + | | |
| | | CG3321 | CG3321 | ATP5I | Subunit e of Fo, dimerization/bending | ++ | | |
| | | CG4307 | Oscp | ATP5O | OSCP subunit of F1/stator arm | ++ | ** | |
| | | CG4412 | ATPsyn-Cf6 | ATP5J | Coupling factor 6, subunit of Fo/stator arm | + | | |
| | | CG6105 | l(2)06225 | ATP5L, ATP5L2 | Subunit g of Fo, dimerization/bending | + | | |
| | | CG7610 | ATPsyn-γ | ATP5C1 | γ Subunit of F1 (stalk) | + | * | |
| | | CG6030 | ATPsyn-d | ATP5H | Subunit d of Fo, stator arm | + | * | |
| | | CG8189 | ATPsyn-b | ATP5F1 | Subunit b of Fo, stator arm | ++ | * | |
| | | CG11154 | ATPsyn-β | ATP5B | Core β subunit of F1 | + | ** | |
| 5 | Mitochondrial biogenesis and dynamics | CG3114 | erect wing | NRF1 | Nuclear respiratory factor 1 homologue | 0 | ** | |
| | | CG6338 | Ets97D | GABPA | Nuclear respiratory factor 2, α subunit | + | ** | |
| | | CG6512 | CG6512 | AFG3L2, SPG7 | m-AAA protease subunit | 0 | * | |
| | | CG8479 | opa1-like | OPA1 | Dynamin-related protein required for inner mt membrane fusion | + | * | |
| | | CG9809 | spargel | PPARGC1A, B, PPRC1 | PPAR γ coactivator | + | ** | Weak positive, diminished number of nucleoid signals remaining in some cells |
| | | CG14981 | maggie | TOMM22 | Tomm 22 subunit of outer mt membrane translocase | + | | |
| 6 | Nuclear gene expression | CG1057 | MED31 | MED31 | Mediator complex subunit 31, trancriptional elongation | 0 | | |
| | | CG1554 | RpII215 | POLR2A | RNA polymerase II 215kD subunit | + | | |
| | | CG1810 | mRNA-cap | RNGTT | mRNA capping enzyme | 0 | | |
| | | CG1874 | Not1 | CNOT1 | CCR4–NOT transcription complex subunit 1 | 0 | ** | Weak positive, diminished number of nucleoid signals remaining in some cells, some cell death also evident |

**Table 1** (Continued)

| Category number | Category name | CG number | Official name or symbol | Human orthologue | Other or common name(s) or putative function | mtDNA depletion | > 3 rescreenings | Note |
|---|---|---|---|---|---|---|---|---|
| | | CG2163 | Pabp2 | PABPN1, PABPN1L | Nuclear poly(A)-binding protein | – | | |
| | | CG3162 | LS2 | | Novel U2AF-related regulator of differential splicing | 0 | | |
| | | CG3675 | Art2 | PRMT6 | Protein arginine methyltransferase 2 | + | | |
| | | CG6525 | pps | SPOCD1, PHF3 | Protein partner of snf, regulator of alternative splicing of Sex lethal | + | * | Diminished number of nucleoids remained in many cells |
| | | CG7626 | Spt5 | SUPT5H | Transcription elongation factor SPT5 (DSIF complex) | 0 | | |
| | | CG9591 | omd | INTS5 | Integrator complex subunit 5, snRNA processing factor | + | * | Very few cells remained, after 5 days of treatment |
| | | CG9748 | belle | DDX3X,Y; DDX4 | RNA helicase, implicated in X-chromosome dosage compensation | 0 | | |
| | | CG10955 | Rtf1 | RTF1 | Component of RNA polymerase II-associated (PAF1) complex, transcriptional elongation | 0 | ** | |
| | | CG11990 | hyrax | CDC73 | Component of RNA polymerase II-associated (PAF1) complex, transcriptional elongation | 0 | * | |
| | | CG17183 | MED30 | MED30 | Mediator complex subunit 30, trancriptional elongation | 0 | ** | |
| | | CG17358 | Taf12 | TAF12 | TATA box-binding protein-associated factor 12 | + | | |
| | | CG17603 | Taf1 | TAF1, TAF1L | TATA box-binding protein-associated factor 1 | + | | |
| 7 | Miscellaneous (other or unknown) | CG3539 | SLY-1 homologue | SCFD1 | ER to Golgi vesicle transport | 0 | ** | |
| | | CG4268 | Pitslre | CDK11A | Cyclin-dependent kinase superfamily, regulator of autophagy | – | | |
| | | CG5794 | CG5794 | USP34 | Ubiquitin-specific peptidase (deubiquitinating enzyme) | ++ | * | |
| | | CG6413 | Dis3 | DIS3 | Exosome complex exoribonuclease | – | * | |
| | | CG7368 | CG7368 | | Zn finger protein | – | ** | |
| | | CG8021 | CG8021 | SLIRP | RNA-binding protein, regulation of mt RNA levels | 0 | | |
| | | CG9007 | upSET | SETD5, MLL5 | Zn finger protein, related to trithorax and histone lysine methyltransferases | – | | |
| | | CG9397-H | jing | AEBP2 | Zn finger transcription factor, role in morphogenesis | – | ** | Was clearly positive in 5 out of 7 trials |

**Table 1** (Continued)

| Category number | Category name | CG number | Official name or symbol | Human orthologue | Other or common name(s) or putative function | mtDNA depletion | > 3 rescreenings | Note |
|---|---|---|---|---|---|---|---|---|
| | | CG9797 | CG9797 | | Zn finger protein | 0 | ** | Weak positive, diminished number of nucleoid signals remaining in some cells |
| | | CG10042 | MBD-R2 | | Methyl-DNA-binding protein | 0 | | Some small nucleoid signals remained in some cells. Another dsRNA for this gene gave consistently negative findings |
| | | CG10144 | CG10144 | VPS8 | Vesicle sorting to lysosomes | + | ** | Nucleoid signals of diminished size or intensity seen in some cells, few cells remained after 5 d |
| | | CG10395 | CG10395 | | Zn finger protein | 0 | | |
| | | CG10582 | Sin | POLR3E | Sex-lethal interactor, possible alternative splicing, proposed subunit of RNA polymerase III | 0 | * | |
| | | CG10582-C | Sin | | Putative mitochondrially targeted isoform of Sin | 0 | ** | Clear positive in 3 trials, but also some inconsistent findings |
| | | CG12242 | GstD5 | | Glutathione S-transferase superfamily | (+) | | Depletion just outside the border of significance, due to large variance |
| | | CG13203-C | CG13203 | | Putative mitochondrially targeted isoform of protein with unknown function | – | ** | |
| | | CG13779 | Sem1 | | Possible endopeptidase | 0 | ** | |
| | | CG14084 | Bet1 | BET1 | ER to Golgi vesicle transport | 0 | * | |
| | | CG14247 | CG14247 | | Unknown function | 0 | * | |
| | | CG14634 | CG14634 | | Unknown | + | ** | |
| | | CG15231 | IM4 | | Immune-induced molecule, peptide hormone? | + | * | |
| | | CG15343 | CG15343 | | Pyridoxamine 5'-phosphate oxidase-like | 0 | * | |
| | | CG15793 | Dsor1 | MAP2K1, 2, 5 | MAP kinase kinase | 0 | * | |
| | | CG17077 | pointed | ETS1, 2 | Ets transcription factor | ++ | * | |
| | | CG31258 | Cenp-C | | Centromere-binding, kinetochore function | nt | * | |

**Table 1** (Continued)

| Category number | Category name | CG number | Official name or symbol | Human orthologue | Other or common name(s) or putative function | mtDNA depletion | > 3 rescreenings | Note |
|---|---|---|---|---|---|---|---|---|
| | | *CG31079* | | | Unknown (gene model later withdrawn from Flybase) | nt | | Predominantly induced cell death, few cells remaining seemed positive |
| | | CG32085 | CG32085 | FBXL16 | Putative ubiquitin ligase | 0 | | |
| | | *CG32561, 2* | xmas-1, xmas-2 | MCM3AP (xmas-2) | Putative protein acetyltransferase involved in DNA replication (xmas-2) | 0 | ** | Clear positive in 4 trials, but also some inconsistent findings |
| | | CG32570 | TwdlY | | Suggested cuticular protein | ++ | * | |
| | | CG32652 | CG32652 | | Unknown | + | * | |
| | | *CG34415* | mute | | Muscle wasted, chromatin protein, histone locus body | 0 | ** | Weak positive, diminished nucleoid signal seen in some cells, cell death seen only in some trials |
| | | *CG33546* | gfzf | | GST-containing Zn finger protein, putative mitotic checkpoint protein | 0 | | |
| | | CG42666-G | CG42666 | REXO1 | Exoribonuclease, predicted mitochondrially targeted isoform | nt | ** | Weak positive, diminished number of nucleoid signals remained in some cells, dsRNA for entire gene was an inconsistent weak positive; see also Table S1. isoform-specific primers also target Adar, CG12598 |
| | | CG42666-D | CG42666 | REXO1 | Exoribonuclease, predicted mitochondrially targeted isoform | 0 | * | Weak positive, diminished number of nucleoid signals remained in some cells, dsRNA for entire gene was an inconsistent weak positive; see also Table S2 |
| | | *CG42281* | bunched | | Signal transduction in response to growth factor (dpp) binding | – | | |

= not in original positives list; = positive isoform with putative mt targeting; = gene model withdrawn from Flybase.
Positives also showing substantial cell death are shown in underline and italic.
> 3 rescreenings: Positives that required many rounds of rescreening for final validation. * = 4 rounds; ** ≥ 8 rounds.

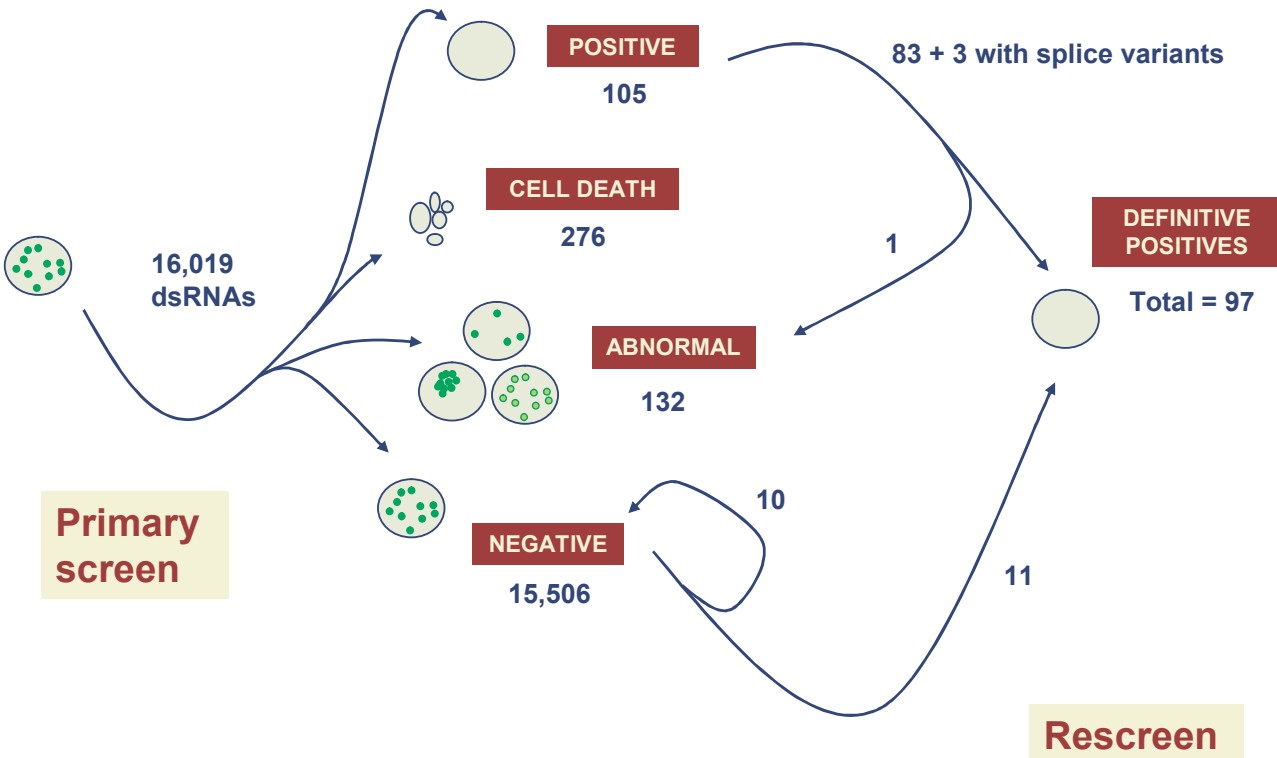

**Figure 2. Overall results of the screen.**
Schematic diagram illustrating number of dsRNAs analyzed, numbers of primary positives, negatives and other classes, and the results of rescreening.

of many genes resulted in loss of PicoGreen signal without major changes in mtDNA copy number suggests that indirect effects may be common, for example, affecting DNA topology, nucleoid architecture, membrane potential or cellular dye uptake.

**Positives implicated in mtDNA metabolism**

The positives include most of the proteins with known roles in mtDNA replication or transcription, notably the five shown previously to be essential for mtDNA maintenance in *Drosophila* (Goto *et al*, 2001; Maier *et al*, 2001; Iyengar *et al*, 2002; Matsushima *et al*, 2004; Humphrey *et al*, 2012). The list comprises the two subunits of PolG, the catalytic subunit of the mitochondrial RNA polymerase, mtSSB, the *Drosophila* homologue of the Twinkle helicase, transcription factors TFAM and mTFB2M, mTERF family members mTTF and mTerf5 (Jõers *et al*, 2013), plus rnh1 (RNaseH1) and belphegor (homologue of mammalian ATAD3). All gave significant mtDNA depletion except for rnh1 and belphegor. DNA ligase III (lig3, CG17227) and mTERF family members mTerf3 and CG15390 (homologue of mammalian MTERF4) were consistently negative.

DNA ligase III was previously reported as dispensable for nuclear DNA repair but essential for mtDNA maintenance in human cells (Ruhanen *et al*, 2011) and mouse (Puebla-Osorio *et al*, 2006; Gao *et al*, 2011). Our data imply that it is redundant to at least one other mtDNA ligase in *Drosophila*. Similar arguments may apply to the absence of any topoisomerase, gyrase, recombinase, resolvase or helicase (other than Twinkle). Our study suggests that few dedicated components of the mtDNA replication apparatus remain to be

identified, but this does not exclude factors with overlapping roles in other cell compartments.

In mammalian mitochondria, the transcriptional apparatus is considered essential for both leading- and lagging-strand synthesis (Clayton, 1982; Fuste *et al*, 2010). RNase H1, also required for mtDNA maintenance in mouse (Cerritelli *et al*, 2003) and human cells (Ruhanen *et al*, 2011), might be involved in primer removal, but this typically also needs other helicases and nucleases such as *Fen1* (CG8648) and *Dna2* (CG2990), both implicated in mtDNA replication in mammalian cells (Duxin *et al*, 2009; Kazak *et al*, 2013). Their absence from the positives list is unsurprising, however, since both also function in the nucleus. One positive from the miscellaneous category, CG8021, appears to be a *Drosophila* homologue of SLRP, a mammalian protein involved in mitochondrial mRNA stabilization and processing (Sasarman *et al*, 2010; Chujo *et al*, 2012).

The mammalian ATAD3 family has been implicated in nucleoid organization (He *et al*, 2007), mitochondrial protein synthesis (He *et al*, 2012), regulation of apoptosis (Huang *et al*, 2011) and autophagy (Chen *et al*, 2011), cholesterol trafficking (Rone *et al*, 2012), mitochondrial dynamics (Gilquin *et al*, 2010) and stress resistance (Hoffmann *et al*, 2012). Loss of PicoGreen nucleoid signal with only a minor drop in mtDNA copy number may indicate that belphegor functions also in diverse pathways and that its effects on nucleoids and mtDNA may be indirect.

Nucleases other than RNase H1 have been shown or suggested to have roles in mtDNA metabolism in various organisms, including EXOG (Tann *et al*, 2011), EndoG (McDermott-Roe *et al*, 2011) and yeast Exo5 (Burgers *et al*, 2010). Exo5 has no *Drosophila*

homologue, and the somatically expressed EndoG/EXOG isogene (EndoG, CG8862) was not detected as a positive, although two other nucleases (CG42666 isoforms D and G and Dis3) appear under the miscellaneous category. Both appear to be targeted to multiple locations in the cell, and, like rnh1, their knockdown did not give rise to significant mtDNA depletion at the 5 days time point. CG42666 has a well-characterized 3′ to 5′ (REX1-like) exoribonuclease domain shared with proofreading DNA polymerases, such as *E. coli* DnaQ (Pol III ε subunit), and with PAN2 deadenylase (Koonin & Deutscher, 1993). Its closest human homologues (REXO1 family) are poorly characterized, thus far implicated only in the suppression of transcriptional pausing (Tamura *et al*, 2003), but REXO2 has recently been reported to function in mitochondria as an oligoribonuclease required for normal mitochondrial morphology, nucleic acid content and protein synthesis (Bruni *et al*, 2013).

Dis3 is predicted to have a mitochondrial localization (Mitoprot II score 97%), supported by other studies (Mamolen, 2010; Hou *et al*, 2012). Its human homologues DIS3 and DIS3L1 are considered to be the main nucleases of the RNA exosome complex (Tomecki *et al*, 2010), with roles in RNA processing, degradation and surveillance in both nucleus and cytoplasm. Analysis of a putative mitochondrial function for CG42666 and Dis3 will thus be problematic, due to multiple localization and pleiotropy.

## Cytosolic translation and the proteasome

These positives include both small and large ribosomal subunit proteins, one translation factor, core and regulatory/subunit components of the proteasome, plus one proteasome assembly chaperone. In total, there are some 91 genes annotated as cytosolic ribosomal proteins in the *Drosophila* genome, plus 7 others as 'ribosomal protein-like'. The frequency of annotated ribosomal proteins in the positives list (14/97, 14%) is thus significantly greater than in the genome as a whole (98/13,937, 0.7%, $P < 0.01$ by $\chi^2$ test). Note also that 38 of 132 genes in the abnormal list (29%, Supplementary Table S3) are annotated as encoding ribosomal proteins, in addition to *Tor* and *raptor*, which positively regulate cytosolic protein synthesis. For proteasomal genes, the calculations are similar (7/97, 7.2%, versus 60/13,937, 0.4%, $P < 0.01$ by $\chi^2$ test).

Knockdown of proteasomal components gave mtDNA depletion mainly in the moderate (+) range, whereas knockdown of some representative cytosolic ribosomal proteins produced either no change in copy number or a relative increase, which may reflect cell death occurring in a large fraction of the cells by the time they were harvested.

One possible explanation for the link between cytosolic translation and mitochondrial homeostasis is that one or more short-lived and/or poorly translated cytosolic mRNAs encode products essential for maintaining mitochondria. Alternatively, mitochondria might be turned over by enhanced autophagy, which results from a prolonged deficiency of protein synthesis (Neufeld, 2012). Consistent with this, Pitslre, a negative regulator of autophagy (Wilkinson *et al*, 2011), was positive in our screen.

A more specific impact on mitochondria could involve mitonuclear regulation to minimize proteotoxic stress due to an insufficient supply of nuclear-coded OXPHOS subunits. The apparatus of mitochondrial translation was absent from the list of positives, consistent with antibiotic studies (Storrie & Attardi, 1972), but in contrast

with findings in fungi, where translational deficiency leads to loss of mtDNA (Contamine & Picard, 2000). However, a stress signal emanating from the mitochondrial ribosome would most logically produce growth arrest (Richter *et al*, 2013), rather than mtDNA depletion.

Concomitant knockdown of mitochondrial and cytosolic ribosomal protein genes produced no rescue of cell-death phenotypes, and no obvious rescue of PicoGreen nucleoid signals (Table 1). Mitonuclear imbalance thus does not seem to explain the identification of cytosolic ribosomal protein genes as positives in the study, despite evidence for its involvement in other phenomena, such as aging (Gomes *et al*, 2013; Houtkooper *et al*, 2013).

The proteasome represents a crucial intracellular system for turnover of damaged or misfolded proteins, and for post-translational regulation. Proteasomal impairment in yeast has previously been shown to entrain mtDNA instability (Malc *et al*, 2009). A possible explanation is that proteasomal insufficiency signals the activation of other turnover pathways impacting mitochondria, for example, via the Pink1/parkin system (Narendra *et al*, 2012). The proteasome may also have more specific targets relevant to mitochondrial homeostasis, as suggested by the identification of a deubiquitinating enzyme (CG5794) as a positive. Two of its yeast homologues, UBP9 and UBP13, regulate mtDNA maintenance via effects on the biosynthesis of ATP synthase (Kanga *et al*, 2012), although the analogy cannot be exact, since the target is mtDNA-encoded in yeast, but nuclear-coded in animals. UBP9/13 and CG5794 are not orthologous, but the underlying principle may be similar. Known targets of ubiquitylation in mitochondria include proteins involved in mitochondrial dynamics, such as mitofusins (Yonashiro *et al*, 2006; Cohen *et al*, 2008). The closest homologues of CG5794 in yeast and human perform diverse tasks related to peroxisomal protein import (Debelyy *et al*, 2011), cell-cycle regulation (Bozza & Zhuang, 2011), methylmercury resistance (Hwang *et al*, 2012), nuclear DNA repair (Sy *et al*, 2013), and signaling (Lui *et al*, 2011; Poalas *et al*, 2013).

## ATP synthase

The primary screen yielded 5 of the 14 nuclear genes encoding subunits of ATP synthase, once again far above random expectation ($P < 0.01$, $\chi^2$ test). After rescreening, 9 ATP synthase genes were classed as definitive positives (Table 1), including subunits of both the $F_O$ and $F_1$ subcomplexes: $F_1$ core subunit β, stalk subunits γ and δ, stator arm subunits b, d, OSCP and Cf6, and $F_O$ subunits e and g, implicated in dimerization. $F_1$ core subunit α and stator arm subunit f were negative, while stalk subunit ε and $F_O$ subunit c were not consistently positive and thus appear in Supplementary Table S4, although may better be considered as weak positives. A putative assembly factor for ATP synthase, the homologue of yeast Atp12, plus the two isogenes (CG13551 and CG34423) for the inhibitor subunit ATPIF1 were negative.

The requirement for ATP synthase (OXPHOS complex V, cV) to maintain normal mtDNA copy number mirrors observations in yeasts (Giraud & Velours, 1997; Lai-Zhang *et al*, 1999; Clark-Walker *et al*, 2000; Contamine & Picard, 2000; Lefebvre-Legendre *et al*, 2003; Duvezin-Caubet *et al*, 2006; Wang *et al*, 2007) and trypanosomes (Schnaufer *et al*, 2005). This suggests the operation of a universal mechanism, whereby cV knockdown generates a signature of bioenergetic stress, signaling a decrease in mitochondrial

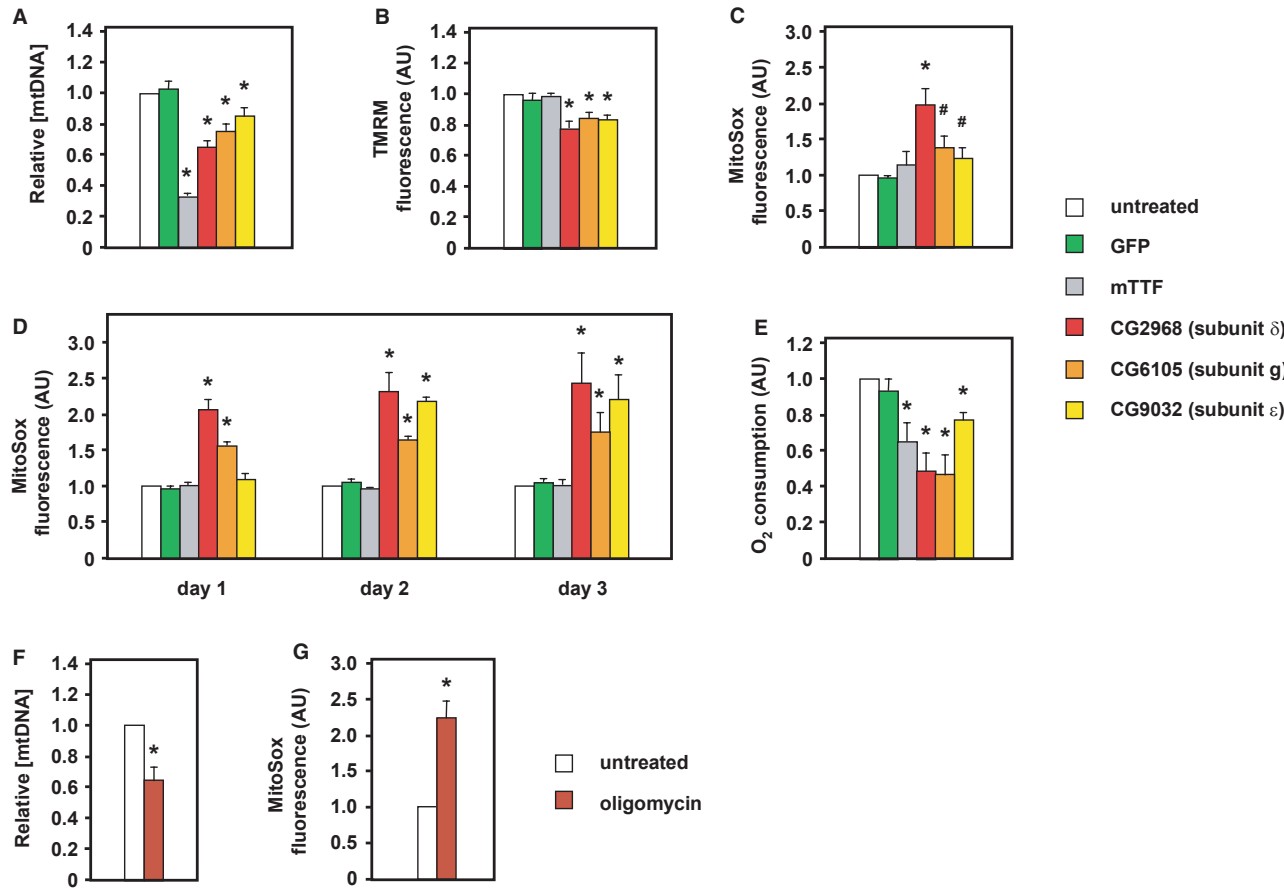

**Figure 3.  Effects of knockdown of cV subunits.**

A–C   Relative mtDNA level (A), TMRM fluorescence (B) and MitoSox fluorescence (C) following treatment for 5 days with dsRNA against the genes indicated.
D     MitoSox fluorescence following treatment for 1–3 days with dsRNA against the genes indicated.
E     Whole cell respiration following treatment for 5 days with dsRNA against the genes indicated.
F, G   Relative mtDNA level (F) and MitoSox fluorescence (G) after 5 days of treatment with oligomycin.

Data information: All data normalized to untreated cells. Note that GFP is a negative control, since the cells have no GFP gene, whereas mTTF is a positive control. Means ± SD from at least 4 independent experiments each conducted in triplicate. Asterisks (*) or hashes (#) indicate significant differences from untreated and from GFP values ($P < 0.01$ and $P < 0.05$, respectively). Note that correction for the decrease in mitochondrial content mitigates the decrease in TMRM fluorescence, but increases still further the MitoSox fluorescence (Supplementary Fig S4D).

Source data are available online for this figure.

biogenesis and/or an increase in mitochondrial turnover. Deficiency of ATP synthase would be expected to disturb mitochondrial membrane potential, impairing the import of nuclear-coded OXPHOS subunits and leading to downregulation of mtDNA as a response to limit proteotoxic stress inside mitochondria. ATP synthase mutations that result in uncoupling are known to lead to mtDNA instability in yeast (Wang *et al*, 2007), and genes encoding subunits of the mitochondrial import machinery can suppress the *petite* negativity of yeast mutants lacking ATP synthase (Dunn & Jensen, 2003).

Another predictable effect of ATP synthase knockdown would be an increase in mitochondrial ROS production, resulting from inhibition of electron flow in the respiratory chain and over-reduction of its various electron carriers. To elucidate the mechanism of mtDNA depletion associated with cV knockdown, we therefore measured mitochondrial membrane potential and ROS production by flow cytometry, using the fluorescent probes TMRM and MitoSox, respectively.

To minimize secondary effects, we selected genes whose knockdown produced moderate mtDNA copy number depletion, confirming first that their relative knockdown potency correlated with the amount of mtDNA depletion produced (Fig 3A, Supplementary Fig S2A). Knockdown of either of the two clear positives (CG6105, subunit g, and CG2968, subunit δ), plus one that gave only a weak knockdown and minimal mtDNA depletion (CG9032, subunit ε), produced coherent effects (Fig 3B and C). In all cases, there was a small but significant decrease in TMRM fluorescence per cell and a more variable increase in MitoSox fluorescence, whereas these parameters were unaffected by knockdown of mTTF, which acts directly on mtDNA transcription (Roberti *et al*, 2006) and replication (Jõers *et al*, 2013).

Based on successive measurements over the first 3 days of dsRNA treatment (Fig 3D), the effect on MitoSox fluorescence also reflected the different potencies of knockdown, with CG2968 (subunit δ) producing the most rapid, substantial and sustained ROS

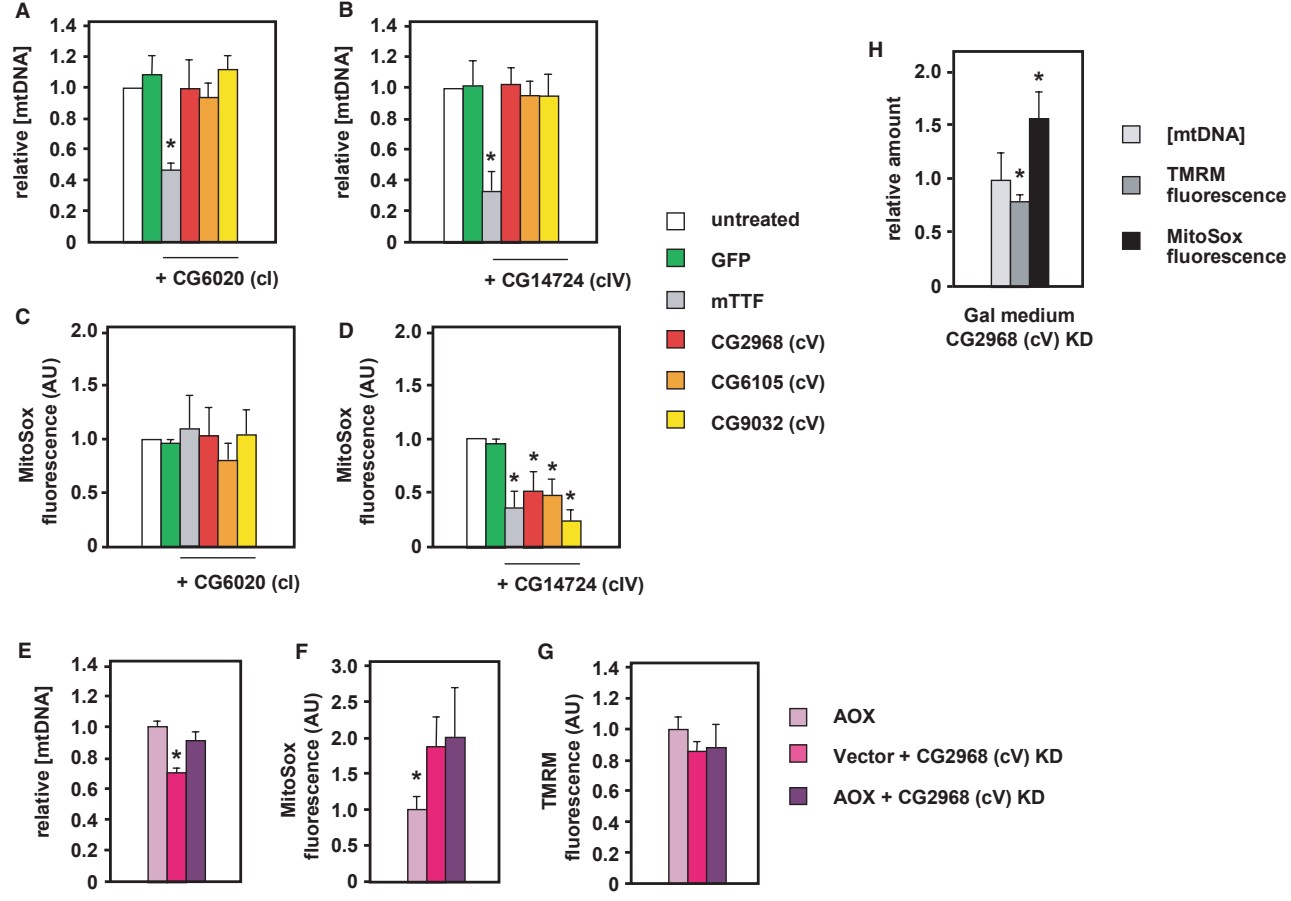

**Figure 4. Effects of knockdown of cV subunits in combination with other treatments.**

A–D   Relative mtDNA level (A, B) and MitoSox fluorescence (C, D) following 5 days of treatment with dsRNA against the genes indicated.

E–G   Relative mtDNA level (E), MitoSox fluorescence (F) and TMRM fluorescence (G) of cells stably expressing AOX or empty vector, as indicated, after 5 days of treatment with dsRNA against CG2968, as shown. Note that these cells were cultured in the presence of hygromycin to maintain the AOX-expressing or control plasmid.

H     Indicated parameters, following 5 days of treatment with dsRNA against CG2968, in cells grown in galactose-containing (Gal) medium.

Data information: All data normalized to untreated cells or (E, F, G) AOX-expressing but otherwise untreated cells grown in parallel. Means ± SD from at least 4 independent experiments, each conducted in triplicate. Asterisks (*) indicate significant differences between experimental and control cell values (P < 0.01).

Source data are available online for this figure.

increase. Knockdown of ATP synthase also decreased whole cell respiration (Fig 3E). Prolonged treatment with a sublethal dose of oligomycin (50 nM, 5 days) phenocopied the effect of cV knockdown on both mtDNA level (Fig 3F) and MitoSox fluorescence (Fig 3G), implicating the enzymatic activity of cV, rather than any structural role in maintaining cristal architecture (Davies *et al*, 2012). Oligomycin binds the *c* ring of cV, most likely at its interface with subunit *a* (Symersky *et al*, 2008), so is unlikely to disturb the dimeric structure of the complex, upon which its membrane-deforming action depends.

We next targeted single-copy genes encoding subunits of the respiratory chain, all of which were negative in the initial screen. When genes for subunits of complex I (CG6020, subunit NDUFA9) or complex IV (CG14724, subunit COX5A) were knocked down individually, there was no significant effect on mtDNA copy number (Supplementary Fig S2B), despite a decrease in TMRM fluorescence (Supplementary Fig S2C) and in whole cell

respiration (Supplementary Fig S2D), similar to that seen after cV knockdown (compare with Fig 3B and E). Knockdown of cI subunit CG6020 produced a small, though non-significant increase in MitoSox fluorescence (Supplementary Fig S2E), while knockdown of cIV subunit CG14724 resulted in a significant *decrease* thereof.

Simultaneous knockdown of subunits of cV and either cI (Fig 4A) or cIV (Fig 4B) abolished the copy number decrease seen when cV alone was targeted (compare with Fig 3A) and prevented the concomitant ROS increase (Fig 4C and D). In contrast, mtDNA depletion by mTTF knockdown was not altered by simultaneous knockdown of cI or cIV subunits. Concomitant knockdown of CG2968 (cV) with genes coding for other subunits of cI or cIV also reversed mtDNA depletion and mitigated ROS increase (Supplementary Fig S3A and B). Knockdown of subunits of cI alone did produce a small drop in mtDNA copy number (Supplementary Fig S3C) and an increase in ROS (Supplementary Fig S3D), though these changes

were not always statistically significant and were comparable with those produced by knockdown of CG9032 (cV subunit ε), thus probably too small for these genes to have been scored as positives in the primary screen.

Overall, these findings indicate increased mitochondrial ROS production as the most consistent marker linking cV knockdown and mtDNA depletion. Accordingly, treatment with the antioxidant TBAP, a chemical mimetic of superoxide dismutase (Faulkner *et al*, 1994), produced a small but statistically significant mitigation of both effects (Supplementary Fig S3E and F), while TBAP alone produced small effects in the opposite direction. However, increased mitochondrial ROS is not sufficient to induce mtDNA depletion.

Stable expression of the *Ciona intestinalis* alternative oxidase (Fig 4E–G), or growth of cells in medium containing galactose in place of glucose (Fig 3H), enforcing the use of OXPHOS to produce ATP (Robinson *et al*, 1992), blocked copy number depletion, but preserved the changes in MitoSox (and TMRM) fluorescence produced upon cV knockdown. Copy number depletion must therefore depend on the integration of different metabolic signals.

To investigate the mechanism of mtDNA depletion, we analyzed mitochondrial turnover, using vital dyes for both mitochondria and lysosomes, employing flow cytometry (Fig 5A) and confocal microscopy (Fig 5B and C). Fluorescence per cell of both LysoTracker Red and NAO, a vital dye for the inner mitochondrial membrane, was

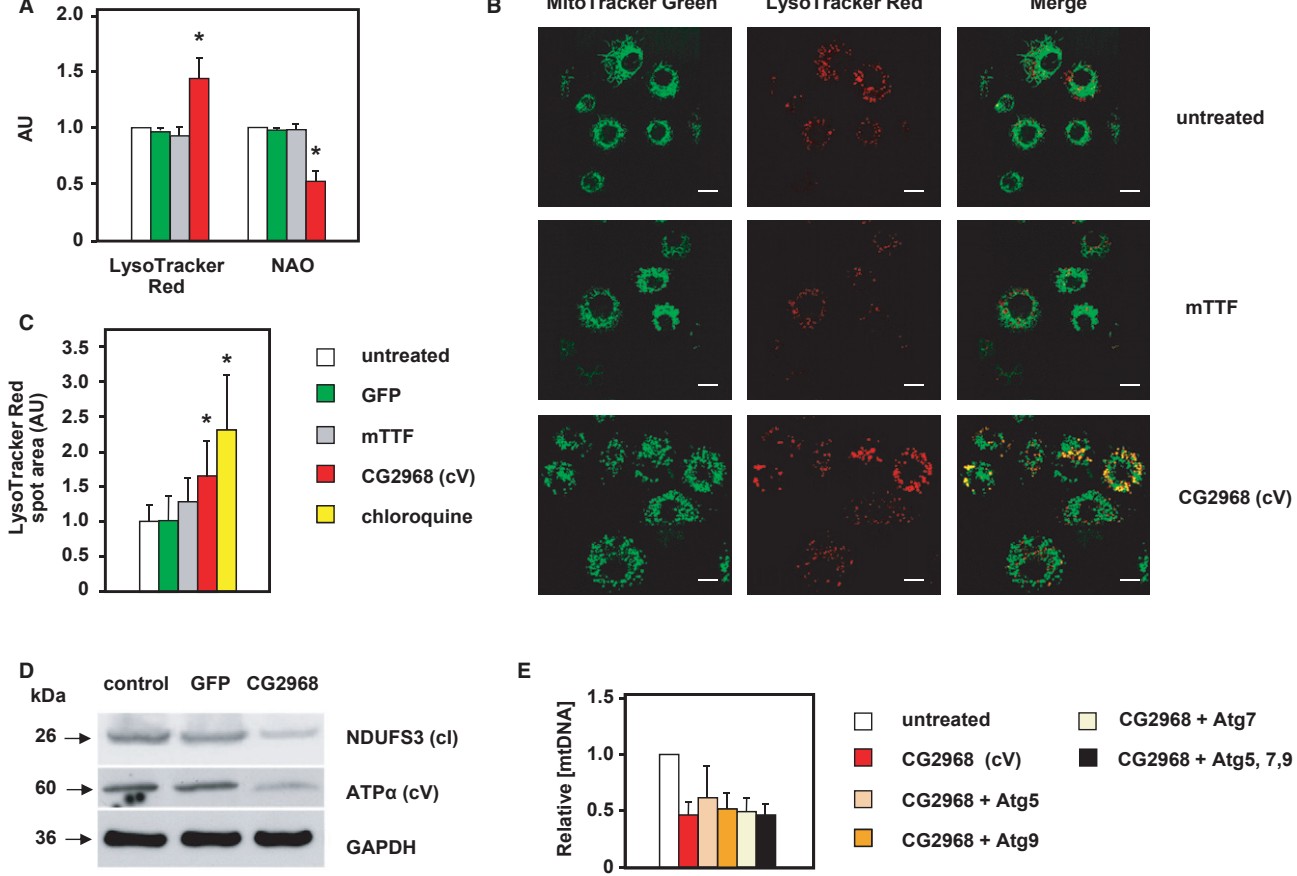

**Figure 5.  Effects of cV knockdown on mitochondrial and lysosomal content.**

A    LysoTracker Red and NAO fluorescence of cells treated as indicated for 5 days with dsRNA against the indicated genes, normalized against the values for untreated cells. Mean ± SD for four independent experiments; asterisks indicate significant differences from untreated, $P < 0.05$.

B    Representative confocal microscopy images of living cells treated for 5 days with dsRNA against the indicated genes and stained with both MitoTracker Green and LysoTracker Red. The scale bar is 10 μm.

C    Mean areas of LysoTracker red staining in cells treated for 5 days with dsRNA against the indicated genes, or with chloroquine. Mean ± SD for three independent experiments, in each of which ≥ 50 cells were analyzed by confocal microscopy in a single plane. Analysis of z-stacks in ≥ 30 cells gave similar results. Note that the increase in lysosome content was intermediate between that of untreated cells and cells treated with chloroquine to block lysosomal turnover of autophagosomes. Asterisks indicate significant differences from untreated, $P < 0.01$.

D    Western blot of total protein extracts from control S2 cells and cells treated with dsRNAs against GFP or CG2968 (cV), probed for NDUFS3 (cI), ATP synthase subunit α (cV) and GAPDH (loading control).

E    Relative mtDNA level after 5 days of treatment with dsRNAs against the indicated genes. Mean ± SD for five independent experiments; other data classes were significantly different from untreated, $P < 0.05$.

Source data are available online for this figure.

unaffected by 5 days of dsRNA treatment against mTTF or GFP, but knockdown of cV (CG2968) increased LysoTracker staining by 50%, while halving the NAO signal (Fig 5A). Western blotting indicated a general decrease in mitochondrial content, affecting cI as well as cV (Fig 5D).

Live imaging confirmed the increase in lysosomal signal per cell (Fig 5B and C); MitoTracker Green staining revealed increased mitochondrial fragmentation and a greatly enhanced colocalization of LysoTracker and MitoTracker signal in most cells (Fig 5B). Concomitant knockdown of three core components of the canonical autophagy machinery was unable to block mtDNA depletion (Fig 5E), indicating that other turnover pathways are involved, such as retro-translocation coupled to proteasomal degradation (Margineantu *et al*, 2007; Heo *et al*, 2010), mitochondrial vesicle delivery to lysosomes, via Pink1/parkin signaling (McLelland *et al*, 2014), or intramitochondrial turnover pathways (see following section).

Co-expression of AOX, which blocked the decrease in mtDNA copy number caused by cV knockdown, attenuated the increase in LysoTracker signal (Supplementary Fig S4A), while chloroquine, which prevents lysosomal acidification, partially rescued mtDNA depletion (Supplementary Fig S4B). These findings are consistent with lysosomal involvement, but do not exclude a contribution from other pathways. Persistent heat stress produced a qualitatively similar effect (Supplementary Fig S4C). Thus, disturbance of the balance between ATP synthase and the respiratory chain complexes may be just one of many processes that produce this outcome. However, given the observations in yeast discussed above, we propose that it represents a conserved homeostatic mechanism for renewal of mitochondria under a variety of metabolic stress conditions.

### Mitochondrial biogenesis and homeostasis

Very few mitochondrial proteins appeared in the 'cell death only' list (Supplementary Table S2), confirming that their broad absence from the main positives list was not due to their being essential for cell survival. Positives that we did find were opa1-like (orthologue of human OPA1 and yeast MGM1), CG6512 (m-AAA protease complex) and maggie (TOMM22), plus three fly homologues of genes for transcriptional activators or co-activators implicated in regulating mitochondrial biogenesis: erect wing (NRF-1), Ets97D (α subunit of NRF-2), and spargel (PGC-1α). Knockdown of these genes typically produced a modest mtDNA depletion, consistent with the idea that mitochondrial content and quality depends on the balance between mitochondrial turnover and biogenesis, which are independently regulated. The role of mitochondrial quality control in pathological states is already well documented (Martinelli & Rugarli, 2007; Ranieri *et al*, 2013; Celardo *et al*, 2014). Clearance of damaged mitochondria is crucial to protection from somatic mtDNA mutations, whereas it may compound defects caused by inherited mutations.

The mitochondrial protein import machinery is required for cell viability because of the essential functions of mitochondria. Therefore, it is surprising that only a single component of this machinery scored positive in the study. Despite extensive functional studies (Shiota *et al*, 2011), there is little to indicate any special property of TOM22 that may explain this finding, although it has been implicated as a site of regulation of the entire TOM complex by cytosolic protein kinases (Schmidt *et al*, 2011). It was recently linked to aging

(Joseph *et al*, 2012) and appears to be a specific target of parkin in promoting mitophagy (Bertolin *et al*, 2013).

Similar questions arise for CG6512. Mammalian OPA1 can be processed by both the m-AAA (Ehses *et al*, 2009) and i-AAA proteases (Song *et al*, 2007), and in *Drosophila* also by the rhomboid protease rho-7 (Rahman & Kylsten, 2011). OPA1 or its isoforms have been previously implicated in mitochondrial genome maintenance (Herlan *et al*, 2003; Sesaki *et al*, 2003; Elachouri *et al*, 2011), although there is no compelling evidence of a direct interaction with mtDNA. The general consensus is that it acts via its documented roles in mitochondrial quality control, affecting cristal morphology and innermembrane fusion (Frezza *et al*, 2006; Meeusen *et al*, 2006), as well as metabolism and calcium homeostasis (Kushnareva *et al*, 2013).

Knockdown of rho-7, CG2658 (paraplegin-like m-AAA component), or CG3499 (i-AAA), all of which were negative in the original screen, produced no significant effects on mtDNA levels (Supplementary Fig S5A). However, we found that these genes, together with opa1-like and CG6512, participate in a mutually interactive network, such that knockdown of any component regulates the others at the RNA level (Supplementary Fig S5B). In particular, knockdown of opa1-like entrained the upregulation of all 4 proteases, while knockdown of CG6512 upregulated CG2658 and CG4399, suggesting a common pathway modulating mitochondrial biogenesis or turnover. Mitochondrial proteases were also upregulated when cV was knocked down.

### Nuclear gene expression

Knockdown of most of the previously characterized components of the apparatus of nuclear gene expression that were positive in the screen (category 6 genes in Table 1) produced no significant change in mtDNA copy number. Since most of them are required for cell survival, a mitochondrial defect may therefore be no more than a staging post on the road to cell death. However, downregulation of six genes, *RpII215*, *Art2*, *pps*, *omd*, *Taf12,* and *Taf1*, with documented or hypothesized roles in chromatin modification, Pol II transcription, or RNA processing, led to decreased mtDNA copy number. A possible role in mtDNA metabolism independent from their functions in the nucleus thus needs to be carefully evaluated.

CG10582 (Sin, Sex-lethal interactor) may also be considered a representative of this group. It was originally identified in a yeast 2-hybrid screen as a partner of the splicing factor Sex lethal (Dong & Bell, 1999), but its orthologues are subunits of RNA polymerase III (Hu *et al*, 2002; Cramer *et al*, 2008), specifically POL3E, the Pol III homologue of TFIIF (Carter & Drouin, 2010; Kassavetis *et al*, 2010), required for termination (Landrieux *et al*, 2006). No other subunits of RNA polymerase III were detected in the screen. One Sin isoform (weakly) predicted to be mitochondrial was also positive in the PicoGreen assay, as was pps (protein partner of snf), also implicated in alternative splicing of Sex lethal (Johnson *et al*, 2010).

pointed (CG17077) has 4 isoforms created by differential splicing, though none is predicted with high confidence to be mitochondrial. It exhibits functional redundancy in specific contexts (Baltzer *et al*, 2009) with its homologue Ets97D (NRF-2α), suggesting that the canonical functions of the latter in regulating mitochondrial biogenesis might be shared by the two transcription factors in *Drosophila*.

**Miscellaneous positives: 'unknowns' and 'others'**

This heterogeneous group mainly comprises poorly characterized proteins. In addition to several putatively implicated in RNA metabolism, plus a number of zinc finger proteins, they include three that function in vesicle trafficking. Bet1 (CG14084) and Slh (CG3539), conserved from yeast to humans, are members of the syntaxin 5-SNARE complex, involved in ER to Golgi transport (Newman *et al*, 1990; Hay *et al*, 1997). Bet1 is a SNARE protein, while Slh, orthologue of mammalian and yeast Sly1 (Dascher *et al*, 1991; Sogaard *et al*, 1994; Dascher & Balch, 1996), is a SNARE-interacting SM-family protein. Knockdown of each produced no significant mtDNA depletion, indicating that loss of PicoGreen nucleoid signal is due to something other than decreased copy number. CG10144 is the *Drosophila* orthologue of yeast (and human) Vps8, involved in vesicle sorting to lysosomes (Chen & Stevens, 1996; Horazdovsky *et al*, 1996). Our findings suggest a novel role for vesicle trafficking in mitochondrial homeostasis.

Aside from these, no specific pathway or cellular complex is identified more than once in the list. Knockdown mainly produced little or no mtDNA depletion, although three further positives that did so warrant further study, namely IM4, previously implicated in innate immunity, CG14634 and CG32652, all of which encode proteins with no identified homologues outside of insects.

## Conclusions

In this study, we use a genome-wide screen to identify novel components of the machinery of mtDNA copy number maintenance and regulation. Most positives belonged to coherent sets, as discussed above (summarized in Fig 6). Unexpectedly, absent from the positives list was any gene involved in mitochondrial protein synthesis or OXPHOS assembly, most known signaling pathways, any subunit of the respiratory chain complexes or of intermediary metabolism, genes involved in handling oxidative stress, or in nucleotide transport and metabolism, which cause mtDNA depletion syndromes in humans (Saada, 2004; Copeland, 2012). The latter is not surprising, since these genes are of pathological importance only in non-proliferating cells, where the standard pathways sustaining the supply of precursors for both nuclear and mtDNA synthesis are unavailable. However, apart from POLG and Twinkle (PEO1), the homologues of most other human mtDNA depletion genes, including MPV17 (Spinazzola *et al*, 2006), FBXL4 (Bonnen *et al*, 2013), and SERAC1 (Sarig *et al*, 2013), were also absent, although some may be false negatives. Others, such as MGME1 (Kornblum *et al*, 2013), have no *Drosophila* homologue.

On the other hand, few novel candidates emerged, for direct involvement in DNA metabolism, despite the burgeoning number of proteins said to be present in nucleoids (Reyes *et al*, 2011;

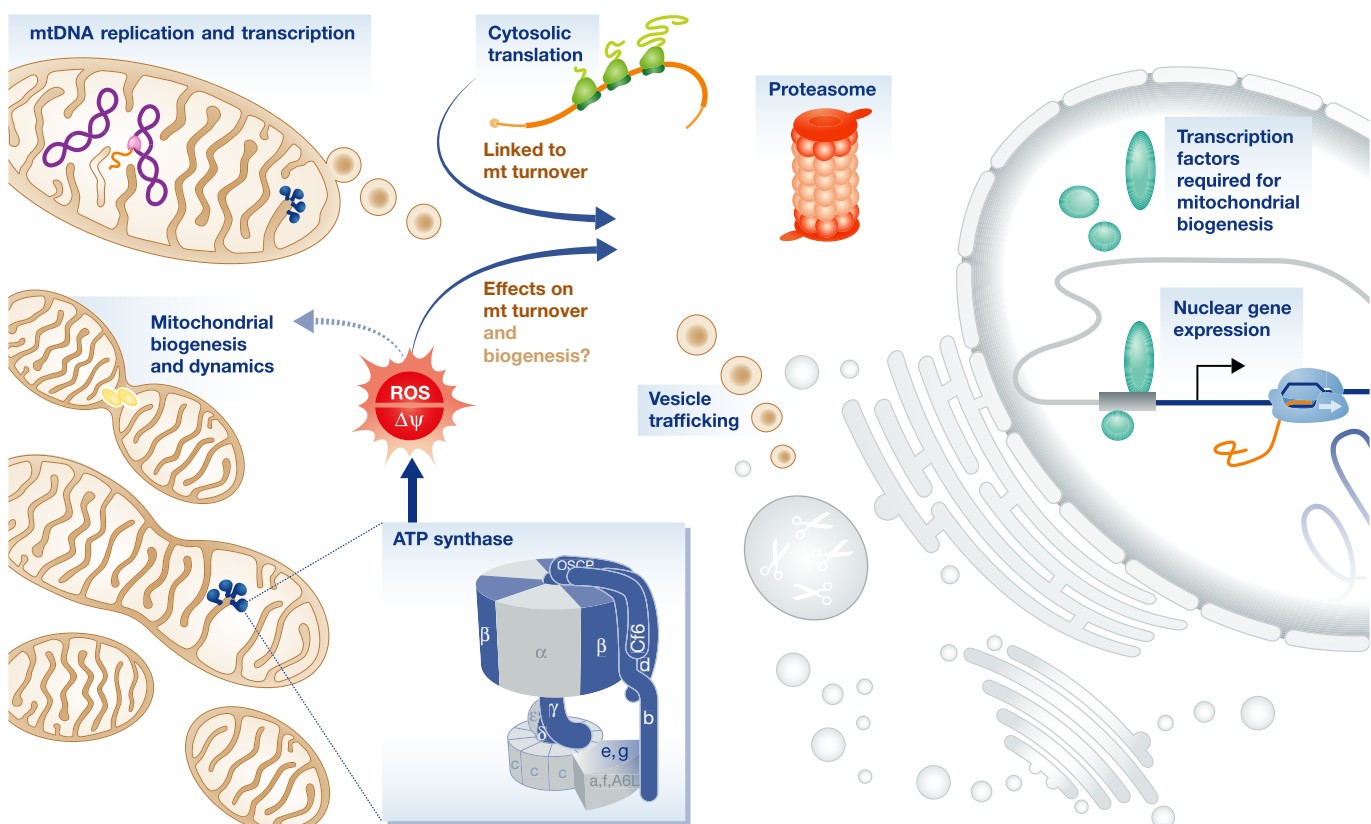

**Figure 6.  Pathways inferred to be involved in mitochondrial DNA copy number maintenance and homeostasis.**
Schematic diagram of the cell, showing categories of identified genes in the positives list (in dark blue), and their possible links to a ROS (or membrane potential, Δψ) surveillance mechanism, orchestrated by ATP synthase.

Bogenhagen, 2012). The involvement of cV, the proteasome, and some key genes for mitochondrial dynamics and quality control suggests that mtDNA copy number is dependent on a balance between mitochondrial turnover and biogenesis, in which specific stresses, notably mitochondrial ROS production and impaired cytosolic protein turnover, may be crucially important.

ATP synthase was inferred to be a key player in homeostatic maintenance of mitochondria, and thus in the amount of mtDNA and its gene products. Mutations in ATP synthase in fungi are already known to play a determining role in whether mtDNA loss can be tolerated (Contamine & Picard, 2000; Lefebvre-Legendre *et al*, 2003), or even facilitated (Giraud & Velours, 1997; Lai-Zhang *et al*, 1999; Contamine & Picard, 2000), but with membrane potential implicated as a key parameter (Duvezin-Caubet *et al*, 2006; Wang *et al*, 2007). In S2 cells, excess ROS production was better correlated with the strength and kinetics of mtDNA depletion, but this could be an epi-phenomenon. The fact that membrane potential 'per mitochondrion' (Supplementary Fig S4D) was restored to its starting value suggests that its disturbance may yet prove to be the primary inducer.

Excess ROS has elsewhere been proposed to lead to mtDNA depletion under pathological conditions (Larosche *et al*, 2010; Quinzii *et al*, 2013), and pathological defects in ATP synthase associated with ROS overproduction (Baracca *et al*, 2007) may downregulate mitochondrial functions (Wojewoda *et al*, 2010, 2011) and even lead to mtDNA loss (Vergani *et al*, 1999; Turner *et al*, 2005). ROS overproduction has been widely suggested both to provoke mtDNA damage, but also to result from it. However, a recent report showed that unrepaired damage leads to mtDNA depletion without increased ROS (Shokolenko *et al*, 2013), and the role of ROS in producing somatic mtDNA mutations in the PolgA mutator mouse is disputed (Trifunovic *et al*, 2005; Dai *et al*, 2010). Thus, the increased mitochondrial ROS seen when cV is knocked down is more logically a cause than a consequence of mtDNA depletion. Furthermore, although ROS may provoke strand breakage, interfering directly with mtDNA replication (Han & Chen, 2013), our data instead suggest that ROS activates mitochondrial turnover before widespread DNA damage would be sustained, as occurs in mammalian cells under TNFα signaling (Nagakawa *et al*, 2005; Vadrot *et al*, 2012). However, an opposing pathway has also been suggested, in which ROS over-production promotes mitochondrial biogenesis, not turnover (Moreno-Loshuertos *et al*, 2006, 2011).

A key aim of future research will be to identify the sensor molecule(s) integrating changes in mitochondrial ROS (or membrane potential) with other metabolic signals, in order to modulate mitochondrial biogenesis and turnover. One possibility consistent with our data is that ATP synthase itself is that sensor.

# Materials and Methods

### Cell maintenance

*Drosophila* S2 cells (Invitrogen) were cultured under standard conditions, in Schneider's medium (Sigma) and diluted 1:6 every 3-4 d. In selected experiments, various drugs were added or glucose was replaced with the same concentration of galactose. S2 cells stably expressing *Ciona intestinalis* AOX were generated by co-transfection with a plasmid conferring hygromycin resistance.

### Screening of *Drosophila* dsRNA library and fluorescence microscopy of nucleoids

S2 cells were seeded into 96-well plates and treated over 5 days with 0.6–1.2 μg of dsRNA from the library (Open BioSystems), alongside positive and negative controls in each plate, as described previously (Jõers *et al*, 2013). Nucleoids were visualized by fluorescence microscopy after staining with Quant-iT™ PicoGreen dsDNA reagent (7.5 μl/ml, Invitrogen). Larger-scale dsRNA treatments for mtDNA copy number evaluation and analysis of cellular parameters were performed essentially as previously (Jõers *et al*, 2013).

### Nucleic acid isolation and QPCR

Total RNA was isolated from S2 cells as previously (Jõers *et al*, 2013). For DNA isolation, cells from a single well of a 24-well plate were processed by a procedure determined to give consistent results irrespective of cell density, involving SDS lysis, proteinase K digestion, isopropanol precipitation and overnight resuspension at 55°C (see SI). For larger-scale experiments, DNA was prepared from $1.5 \times 10^6$ cells cultured in 6-well plates, as previously (Jõers *et al*, 2013). Mitochondrial DNA copy number was assessed by QPCR using primers against COXII or 16S rRNA (for mtDNA) and RpL32 (nuclear DNA, single-copy, for normalization). Transcript levels were estimated relative to that of RpL32 by a similar procedure, but using cDNA as template.

### Measurements of mitochondrial function

Mitochondrial membrane potential, ROS level, and content per cell were determined by flow cytometry (Cannino *et al*, 2012) of cells stained, respectively, with 200 mM tetramethylrhodamine methyl ester (TMRM), 2.5 μM MitoSox™ (Invitrogen), or either 200 nM 10-nonyl acridine orange (NAO) or 40 nM MitoTracker Green FM (Life Technologies). Oxygen consumption of living cells (Cannino *et al*, 2012) was measured using a Clark-type electrode (Hansatech Oxyterm system).

### Analyses of lysosomal and mitochondrial content

Lysosome content per cell was measured by flow cytometry of cells stained with 50 nM LysoTracker Red DND-99 (Life Technologies). MitoTracker Green FM (Molecular Probes) and LysoTracker Red DND-99 were used for live cell imaging by confocal microscopy of mitochondria and lysosomes, respectively, with spot-area calculation (ImageJ) and image deconvolution (SVI, Huygens software).

### Western blotting

Post-nuclear extracts resolved by SDS-PAGE were electroblotted and probed using standard methods (essentially as Fernandez-Ayala *et al*, 2009; see SI). Primary antibodies used were against NDUFS3 (Abcam, mouse), 1:10,000), ATP5A (Abcam, mouse, 1:1,000), and GAPDH (Everest Biotech, goat, 1:2,000), with appropriate horseradish peroxidase-conjugated secondary antibodies. Visualization

used the ECL system (Amersham Biosciences) according to the manufacturer's protocols.

### Statistical analyses

Comparisons between populations were performed using unpaired two-tailed Student's *t*-tests or analyses of variance when more than two samples were compared, with Bonferroni-corrected post hoc *t*-test.
    For further details, see Supplementary Materials and Methods.

### Data availability

Original images from the primary screen are deposited at: http://dx.doi.org/10.5061/dryad.v55p5.

**Supplementary information for** this article is available online: http://msb.embopress.org

### Acknowledgements

This work was supported by funding from the Academy of Finland, Tampere University Hospital Medical Research Fund, and the Sigrid Juselius Foundation. AF was supported by FY 2008 Researcher Exchange Program between JSPS and Academy of Finland. We thank Tea Tuomela, Outi Kurronen, Hanna Ojala, and Eveliina Kaulio for technical assistance, and Susanna Valanne, Mika Rämet, Ian Holt, Cory Dunn, Brendan Battersby, Anu Suomalainen, and Laurie Kaguni for useful discussions and advice.

### Author contributions

AF initiated the project, conducted the primary screen, and supervised the implementation and interpretation of the secondary screening. GC conducted the functional analysis of ATP synthase knockdown. MG performed the copy number analysis and profiled the expression of mitochondrial proteases and opa1-like at the RNA level. SB and SK assisted technically with copy number analysis. EL and AR devised the computational tool for objectifying the Pico-Green fluorescence signals (details to be published elsewhere). FS generated the AOX-expressing S2 cells. ED provided guidance and input for ATP synthase experiments. HTJ devised the project, analyzed the data, compiled the figures and tables, and wrote the text.

### Conflict of interest

The authors declare that they have no conflict of interest.

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
