## [Review Process File · Molecular Systems Biology]

Screen for mitochondrial DNA copy-number maintenance genes reveals essential role for ATP synthase

Atsushi Fukuoh, Giuseppe Cannino, Mike Gerards, Suzanne Buckley, Selena Kaziancioglu, Filippo Scialo, Eero Lihavainen, Andre Ribeiro, Eric Dufour, Howard T. Jacobs

Corresponding author: Howard T. Jacobs, University of Tampere

Review timeline:	Submission date:	12 January 2014
	Editorial Decision:	06 February 2014
	Revision received:	03 April 2014
	Editorial Decision:	27 April 2014
	Revision received:	02 May 2014
	Accepted:	02 May 2014

Editor: Thomas Lemberger

Transaction Report:

1st Editorial Decision

06 February 2014

Thank you again for submitting your work to Molecular Systems Biology. We have now heard back from the three referees who agreed to evaluate your manuscript. As you will see from the reports below, the referees find the topic of your study of potential interest. They raise, however, several concerns on your work, which should be addressed in a revision of the present manuscript.

The major comments raised by the reviewers refer to the following points:

- The need to describe better the global characteristics of the screen (eg in terms of false positive and false negative rates, distribution of the hit scores).
- The need for some additional validation of the model explaining the link between complex V and mtDNA depletion. The points raised by reviewer #1 and #3 seem relevant in this regard. The test of antioxidants suggested by reviewer #2 seems in fact to be already addressed in the paper (TBAP experiment).

We note that the reviewers also suggest some further experimentation to examine the impact of mitochondrial translation proteins (reviewer #3), or of the nucleotide salvage pathway. While we encourage you to include these results if possible, we do not consider that these further reaching data are a condition for publication.

On a more editorial level, we would kindly ask you to address the following points:

- The structure of the text should be clarified and simplified. We copy below the Table of Content of the current manuscript that highlights, to some extent, the issue in terms of complexity of the

structure and possible sources of redundancy. One possible difficulty is that the pathways identified in the screen are first described and then re-discussed more extensively, with interspersed sections reporting the results and discussion of the complex V story.

- We would strongly recommend to shorten and clarify the title.
- To provide a more intuitive view of the results of the screen, we would also recommend including a summary figure depicting the pathways identified with the screen
- With regard to accessibility to the data, we would ask you to include in Materials & Methods a link to the full imaging dataset.

If you feel you can satisfactorily deal with these points and those listed by the referees, you may wish to submit a revised version of your manuscript. Please attach a covering letter giving details of the way in which you have handled each of the points raised by the referees.

REFeree REPORTS:

Reviewer #1:

The authors report a *Drosophila*, genome-wide RNAi screen for regulators of mtDNA copy number. The paper comes from one of the leading authorities on the subject of mtDNA molecular biology. The screen identifies a focused set of pathways, including some positive controls. The most striking result is the pile-up of hits corresponding to complex V. The authors suggest a link between complex V, mitochondrial quality control, and mtDNA copy number.

Major Points:

- (1) It is not clear to the reader what the primary screen was. How was picogreen quantified? Manually, by eye? Automatically, by a computer? How reproducible is this assay? How does it perform on control RNAi versus positive control RNAi? These panels should be shown to inspire confidence in the assay.
- (2) The paper would benefit from a main text, "schematic overview" figure that shows the flow of the screen, beginning with the number of genes tested, the primary assay and criteria for scoring in it, number of genes passing the primary assay, the secondary assay, number of genes passing the secondary assay, etc. For example, there is a reference to RNAis that cause massive death, so were two screens performed in parallel, or in series?
- (3) Certainly some of the positive controls were recovered, but was this greater than chance alone? What was the distribution of the primary screening score over all genes/RNAi tested (a histogram)? What was the distribution of negative controls (random hairpins), and what was the distribution of positive controls (which ought to have been defined in advance), from the screen? Can the authors motivate a "cutoff" on the basis of the positive and negative controls, with which it would be possible to ascribe a false discovery rate for each score?
- (4) The categories of genes that are recovered (cytosolic ribosome, proteasome, complex V) also happen to be some of the most abundant macromolecular machines in the cell. Could the authors plot the primary assay score as a function of estimated abundance of targeted gene product (available from public microarrays) to determine if this is a systematic confounder of the screen.
- (5) It would be very interesting to see what the distribution of the primary screening score was for all of the human homologs whose mutation is associated with mtDNA depletion syndromes.
- (6) Do the authors see any RNAis that increase the mtDNA content per cell, even after correcting for effects on cell size?
- (7) The most interesting part of the paper -- and indeed the title focus -- is the link between complex

V and mtDNA. Do oligomycin (or other inhibitors of complex V) impact mtDNA copy number? What is the impact of ATPIF1, the inhibitory subunit? The work of David Mueller (mgi mutations) ought to be mentioned and considered here.

(8) I think the manuscript needs to be re-written for clarity. It's very difficult to parse -- as written and as the figures are presented. I think with a serious re-write and inclusion of key new figures, the important results can be highlighted while the study's limitations are clearly conveyed.

Reviewer #2:

Fukuoh et al. performed the genome-wide RNAi screening for genes involved in mtDNA maintenance in *Drosophila* S2 cells. The authors noticed that short term RNAi of TFAM rather enhanced the PicoGreen staining of mtDNA nucleoids. Taking advantage of this phenomenon, the authors set up a unique assay system for monitoring fluorescence of mtDNA nucleoids stained with PicoGreen after dsRNA treatment together with a dsRNA directed against TFAM. The authors present a list of "positive" genes, the knockdown of which decreased the PicoGreen fluorescence. The list contains most genes which are already known to be required for mtDNA metabolism, enforcing the reliability of the list. The authors discuss the genes extensively and systematically citing many references, which increases the usefulness of this database. Thus, this database will surely provide important information for studying the mechanism of mtDNA metabolism where many points are still required to be elucidated.

Specific points

(1) In many of the positive genes, the loss of the PicoGreen staining does not lead to a decrease in the mtDNA copy number. In analogy to ATAD3, the authors simply consider that they change mtDNA structure. However, considering ATAD3's function is related to the mitochondrial translation and so its effects on the PicoGreen staining could be indirect, there is a good possibility that many of the above genes are not directly involved in the mtDNA metabolism.

(2) Many subunit genes of the ATP synthase, i.e. complex V, are positive. This is a promising finding. The authors raise an interesting and unexpected hypothesis that the ATP synthase may be a sensor for integrity of mtDNA and/or mitochondrial turnover. The experimental characterization in this work is limited to the ATP synthase among the over 100 positive genes. However, its characterization is disappointingly circumstantial. No direct evidence is presented. For example, if the authors consider ROS is involved, at least effects of antioxidants should be examined when a gene of the ATP synthase is knocked down.

Reviewer #3:

This is an interesting and important study that addresses the regulation of mtDNA copy number. The authors perform a genome-wide RNAi screen for genes that result in decreased mtDNA (based on pico green staining) in *Drosophila* S2 cells when reduced in expression. The study is performed very well and the results are interpreted very carefully and conservatively, as is appropriate. In the end, not only are proteins that have known roles in mtDNA metabolism confirmed, but new factors and pathways are identified that will spur future studies by the field and lead to greater understanding of mitochondrial homeostasis. The authors focus in mechanistically some on the interesting observation that knock-down of many subunits of ATP synthase (but not other OXPHOS complexes) leads to mtDNA depletion and ROS production and put forth a model that loss of complex V leads to autophagy-mediated degradation that causes the observed depletion. I have a few suggestion to improve the study which are detailed below:

1. That cytoplasmic translation genes were identified in the screen, but not any involved in mitochondrial translation is interesting. In the case of known transcription and replication proteins the authors went back and tested a few known components that were not identified in the screen (e.g. TFB2M) and found they were positive for mtDNA depletion. It is worth testing a few mitochondrial translation proteins directly in the same way to confirm there is no effect on mtDNA copy number. MRPL12, for example, could be tested given it has a role in mitochondrial

transcription in mammalian cells and has other roles in cell cycle, etc. in *Drosophila*. This, and a couple of other MRPs might be revealing. In this regard, it is noteworthy that recent studies have implicated an imbalance in mtDNA versus nuclear-encoded OXPHOS subunits as causing mitochondrial stress (see recent *Cell* paper by Sinclair and *Nature* paper by Auwerx). This might provide some insight into why decreased cytoplasmic translation specifically is causing mtDNA depletion that is worth some discussion.

2. The authors proposed model that complex V knockdown leads to mtDNA depletion due to increased autophagy is based merely on correlation with increased markers of autophagy. This model could easily be tested by simultaneously knocking-down autophagy by RNAi and rescuing the depletion.

3. Another class of proteins that was surprising not to see identified in the screen is ribonucleotide reductase and/or dNTP salvage pathways, which have been implicated in mtDNA copy number regulation in yeast and/or mtDNA depletion syndromes in humans. Can the authors test this directly as well to see if regulation at this level does not exist in *Drosophila*, or if they were just missed in the screen. Maybe the fact that TFAM had to be simultaneously knocked down to facilitate the screen precluded identifying these factors?

Minor points:

- Page 10, misplaced period after "mtTFB2M." Probably suffices at this point to just call this protein "TFB2M"
- Page 12 "comprizes" is misspelled
- Page 28. That PolG-mutator mouse does not cause ROS is controversial (e.g. phenotypes are rescued by targeting catalase in mitochondria). May want to restate or remove statements along these lines.
- Recently Tel1 (yeast) and its homolog ATM in humans have been implicating in sensing ROS and modulating mitochondrial function. This could be discussed.
- The specificity of MitoSox for superoxide is controversial, might be better to just say increased ROS.

The title really does not make sense. Maybe "Screening for factors required for mtDNA maintenance reveals a role for ATP synthase and non-mitochondrial cellular components"

Response to Editor and Reviewers

Editor

In our revised version we have addressed the major criticisms of the reviewers. Relevant changes are shown in red text, and numbered by marginal red boxes numbered as here (A, B, C referring to comments of reviewers 1, 2 and 3, respectively, E referring to the specific comments of the editor).

E1. Need to describe better the global characteristics of the screen

We have tried as best we can to do this without getting mired in detail. The main point is simple: 18% of the primary positives did not rescreen, so the false positive rate in the primary screen was 18%. Based on rescreening of primary negatives in specific classes and a few at random, we estimate that we have failed to find at least as many additional positives as we did find, so the false negative rate is about 100 in 16,000, or a little under 1%. We now make these points in the concluding section on the screen, which we have simplified and shortened.

E2. Additional validation of link between complex V and mtDNA depletion

We have conducted the suggested experiments and include the new findings in the manuscript (see specific points in response to reviewers, below).

E3. Further experiments on mitochondrial translation proteins

We have conducted the suggested experiments and include the new findings in the manuscript, even though they add little (see specific points in response to reviewers, below).

E4. Simplification of the structure of the manuscript

We dispensed with the more historical style and covered all aspects of the screen in the first section, and then the subclasses of genes that we found in successive, dedicated subsections. The complex V story is presented in full, at its appropriate place in the list. The manuscript is thus streamlined, shortened and more logical.

E5. Request to shorten and clarify title

This we found hard. The suggestion of reviewer 3 is longer and even more cumbersome. We accept that some non-native readers might have difficulties with our word order, which is, nevertheless, grammatically correct. We feel the paper will attract wide attention only if the systematic screen and complex V findings are somehow both mentioned. Distilling it down to just 'Essential role for ATP synthase in mitochondrial DNA copy number maintenance' may not fit well to the mission of MSB, and does not mention the broader screen. If you feel it to be more appropriate, we would be happy to alter the title to something more generic and systematic, such as 'An RNAi screen for proteins required for mitochondrial DNA copy number maintenance' but would prefer to retain mention of cV if possible. We leave the final decision to you, and would be happy to discuss further should you wish.

E6. Summary figure

We now include the suggested summary figure depicting the pathways identified with the screen (new Fig. 6). Should MSB wish to redraw it in house-style we would of course be happy. I am not much of an artist. Many additional arrows could have been included, but almost all of them would be speculative, and would make the figure unnecessarily busy and complex.

E7. Data accessibility

We now include the dataset on all the definitive positives as a supplementary file. To aid discoverability we think it is impractical to build a full repository of images of all the negative primary data, since if anyone thinks we have overlooked a positive, of which undoubtedly there are many, they will anyway need to implement their own test rather than take our own data as the last word. We therefore created an excel file, listing the primary screen outcome for every target in the library, as an additional supplementary data file (new Table S1). The lists of primers and dsRNAs they create, provided by the manufacturer, are supplied as additional source data.

Reviewer #1

A1 Clarification of details of the screen

We are a little perplexed, since the answers to most of the reviewer's questions were already in the manuscript, and the crucial question of discrimination was illustrated by examples. In shortening the manuscript we have attempted to draw attention to the answers to these points. The new Figure S1 presents the full data on all the positives.

A2. Schematic overview Figure

A figure as suggested is now included (new Fig. 2).

A3. Cutoffs

The screen was scored as 'all or none', therefore a histogram and estimates of FDR would be meaningless. We can, however, extrapolate the frequencies of false positives and negatives, which are included. The number of dsRNAs shown previously to be required for mtDNA maintenance in *Drosophila* is very small (TFAM, mtTFB2M, mtSSB and the two subunits of Polg, are the only examples to our knowledge, and all were positive in the blinded screen). mTTF and mTerf5, recently published by ourselves, actually came out of the screen. A statistical analysis on such low numbers would therefore be meaningless. The only negative controls we tested were GFP and a blank, in every plate. We also included these controls in the QPCR assay. Based on over 40 repeats there was no significant change in mtDNA copy number compared with control cells.

A4. Significance of observed recovery of key gene categories

The recovery of cytosolic ribosomal proteins, proteasomal subunits and ATP synthase subunits is far above expectation, with high significance (χ^2 test, $p < 0.01$ in each case), as now detailed in the manuscript.

A5. Homologues of human mtDNA depletion genes

Apart from POLG and Twinkle (PEO1), such genes were not identified in our screen. We added a note specifying this. See also point C3, below.

A6. Increased copy number

As already detailed in Table 1, we did find a few examples of increased copy number, based on QPCR, although we did not attempt to correct for cell size. Such studies are, in fact, under way for a specific set of positives, and will be reported in a future publication.

A7. Oligomycin, ATP1F1, *mg1* mutations

We performed additional experiments as suggested. Prolonged treatment with a sublethal concentration of oligomycin produced the same effect as knockdown of cV subunits, indicating that the enzymatic activity of cV is crucial (see new panels F, G of new Fig. 3). We have added a note that ATP1F1 is encoded in *Drosophila* by two isogenes (CG13551 and CG34423, formerly CG11079), which may be functionally redundant. Both were negative in the screen (see new Table S1). Mueller's work on the *mg1* mutations was already referenced, but its relevance is now further emphasized.

A8. Clarity

We have re-written the manuscript in a more logical format, with the illustrative figures suggested. We hope this improves clarity and readability.

Reviewer #2

B1. Indirect effects

We agree with the reviewer's comment, and have indicated this specifically in regard to the ATAD3 homologue *belphegor*, as well as more broadly in the section on copy number measurements.

B2. Antioxidants

Treatment with the antioxidant TBAP at the same time as ATP synthase knockdown was already included in the manuscript (Fig. S2E, F), and produced data consistent with the ROS hypothesis.

Reviewer #3

C1. Mitochondrial ribosomal proteins

We tested whether concomitant knockdown of several MRPs (mRpL12, *tko*, *bonsai*) could reverse the effects of cytosolic ribosomal protein knockdown, but found no evidence of any rescue. We agree that the data from Sinclair and Auwerx are intriguing, but not obviously relevant here.

C2. Autophagy

In fact, we were careful throughout the original manuscript not to implicate autophagy as such, since we had no evidence that this, rather than some other turnover pathway, is involved in mtDNA depletion upon cV knockdown. However, as suggested by the reviewer, we now tested this directly by concomitantly knocking down genes for 3 core components of the canonical autophagy machinery, Atg5, 7 and 9. Unambiguously, the results indicate that the canonical autophagy machinery is not required, and thus implicate other turnover pathways, such as the retrotranslocation/proteasomal pathway studied by Rutter and colleagues, the vesicle and lysosome-mediated pathway studied by Fon, McBride and colleagues, and/or intramitochondrial pathways (supported by our own data in Fig. S5B). The fact that chloroquine partially blocks the effect is consistent with lysosomal involvement, but doesn't exclude that other pathways are also mobilized.

C3. Ribonucleotide reductase and dNTP salvage pathways

Their involvement in mtDNA depletion syndromes in humans is associated with dysfunction of post-mitotic cells. In actively dividing cells these pathways are minor, and other pathways supply precursors for both mitochondrial and nuclear DNA synthesis. Thus, their absence from the list of positives in the present study is to be expected. We did, however, test whether concomitant knockdown of TFAM might be required for copy number depletion upon cytosolic ribosomal protein knockdown, but found no evidence of such an effect (we have omitted this negative data from the revised paper because it adds nothing).

C4. TFB2M

We corrected the typos. mtTFB2M is the official gene name in *Drosophila*.

C5. PolG mutator mouse

We agree that the role of ROS is controversial, and have acknowledged this, including the catalase reference.

C6. Tell and ATM

The paper is already long, so we omitted this and many other negatives which could also be discussed. Since we did not exhaustively rescreen the negatives, there is little value in discussing isolated genes that were not identified as positives in the study.

C7. MitoSox and superoxide

We acknowledge the controversy about what MitoSox detects, and have therefore expunged all reference to superoxide, in favour of the more generic ROS.

C8 Title

See above, comment E5.

Thank you again for submitting your work to Molecular Systems Biology. We are now globally satisfied with the modifications made and we will be able to accept your manuscript for publication in Molecular Systems Biology pending the following minor points:

- please move the Materials & Methods section after the Discussion section.
- please add a Data Availability section at the end of the Materials & Methods Section that provides a link to the imaging dataset or to the Supplementary dataset (if this is possible in view of the size of the dataset). A possible external repository for such datasets is Dryad (see example: <http://datadryad.org/resource/doi:10.5061/dryad.35h8v>).